# Burden of lower respiratory infections and associated risk factors across regions in Ethiopia: a subnational analysis of the Global Burden of Diseases 2019 study

Amanuel Yigezu ® ,[1] Awoke Misganaw ® ,[1,2] Fentabil Getnet,[1,3]
Tezera Moshago Berheto,[1] Ally Walker,[2] Ababi Zergaw,[1,4] Firehiwot Abebe Gobena,[5]
Muluken Argaw Haile,[5] Alemayehu Hailu ® ,[6] Solomon Tessema Memirie ® ,[7,8]
Dereje Mengistu Tolosa,[5] Semagn Mekonnen Abate ® ,[9] Mesafint Molla Adane,[10]
Gizachew Taddesse Akalu,[11,12] Addis Aklilu,[13] Dejen Tsegaye ® ,[14] Zeleke Gebru,[15]
Mulusew Andualem Asemahagn ® ,[16] Daniel Atlaw ® ,[17] Tewachew Awoke,[18]
Hunegnaw Abebe,[19] Niguss Cherie Bekele,[19] Melaku Ashagrie Belete ® ,[20]
Tekleberhan Hailemariam,[21] Alemeshet Yirga,[22] Setognal Aychiluhm Birara ® ,[23]
Belay Boda Abule Bodicha,[13] Chuchu Churko ® ,[15] Feleke Mekonnen Demeke,[10]
Abebaw Alemayehu Desta,[24] Lankamo Ena,[13] Tahir Eyayu ® ,[25]
Zinabu Fentaw ® ,[26] Daniel Baza Gargamo,[27] Mesfin Damtew Gebrehiwot,[20]
Mathewos Alemu Gebremichael,[13] Melaku Getachew,[28] Getahun Molla ® ,[29]
Biniyam Sahiledengle ® ,[30] Bereket Beyene,[13] Migbar Sibhat ® ,[31]
Negussie Boti Sidamo,[15] Damtew Solomon,[32] Yonatan Solomon ® ,[33]
Birhanu Wagaye,[1,20] Shambel Wedajo ® ,[20] Melat Weldemariam,[34]
Yazachew Engida Yismaw ® ,[18] Moshen Naghavi[35]

AY and AM are joint first authors.

For numbered affiliations see end of article.

**Correspondence to**
Amanuel Yigezu;
yigezuamanuel@yahoo.com

## ABSTRACT

**Objective** This analysis is to present the burden and trends of morbidity and mortality due to lower respiratory infections (LRIs), their contributing risk factors, and the disparity across administrative regions and cities from 1990 to 2019.

**Design** This analysis used Global Burden of Disease 2019 framework to estimate morbidity and mortality outcomes of LRI and its contributing risk factors. The Global Burden of Disease study uses all available data sources and Cause of Death Ensemble model to estimate deaths from LRI and a meta-regression disease modelling technique to estimate LRI non-fatal outcomes with 95% uncertainty intervals (UI).

**Study setting** The study includes nine region states and two chartered cities of Ethiopia.

**Outcome measures** We calculated incidence, death and years of life lost (YLLs) due to LRIs and contributing risk factors using all accessible data sources. We calculated 95% UIs for the point estimates.

**Results** In 2019, LRIs incidence, death and YLLs among all age groups were 8313.7 (95% UI 7757.6–8918), 59.4 (95% UI 49.8–71.4) and 2404.5 (95% UI 2059.4–2833.3) per 100 000 people, respectively. From 1990, the corresponding decline rates were 39%, 61% and 76%, respectively. Children under the age of 5 years account for 20% of episodes, 42% of mortalities and 70% of the YLL

### STRENGTHS AND LIMITATIONS OF THIS STUDY

⇒ The analysis has considered political, government and administrative changes of regional states and cities over the years to map available data and populations.

⇒ The analysis used all available data identified through an extensive collaboration effort and involving more than 700 leading researchers and policy-makers from Ethiopia.

⇒ When data were not available for a particular regional states or city, the modelling process used data from other locations borrowing strength from geographical locations and time, and use predictive covariates.

⇒ However, limited quality data availability and accessibility for the analysis resulted in wider 95% uncertainty intervals which largely affect policy debates, prioritisation and health decisions.

of the total burden of LRIs in 2019. The mortality rate was significantly higher in predominantly pastoralist regions—Benishangul-Gumuz 101.8 (95% UI 84.0–121.7) and Afar 103.7 (95% UI 86.6–122.6). The Somali region showed the least decline in mortality rates. More than three-fourths of under-5 child deaths due to LRIs were attributed to

malnutrition. Household air pollution from solid fuel attributed to nearly half of the risk factors for all age mortalities due to LRIs in the country. **Conclusion** In Ethiopia, LRIs have reduced significantly across the regions over the years (except in elders), however, are still the third-leading cause of mortality, disproportionately affecting children younger than 5 years old and predominantly pastoralist regions. Interventions need to consider leading risk factors, targeted age groups and pastoralist and cross-border communities.

## INTRODUCTION

Lower respiratory infections (LRIs) have been a predominant health problem worldwide, causing more than 2.3 million deaths in 2016 alone, amounting to a mortality rate of 32.2 per 100 000 people.[1] LRIs comprise diseases of the lower airways such as pneumonia, bronchitis and bronchiolitis, among others.[2] Nearly all (99%) of LRI deaths occur in low-income and middle-income countries and highly affect children under the age of 5 years.[3] In sub-Sahara African countries, the mortality rate is 66.4 per 100 000 people, which is four times the mortality rate in East Asia and twice the global average.[2]

Ethiopia ranked in the top three African countries in the number of under-5 child deaths from LRIs.[4] To prevent child death from LRIs and other diseases in early life, Ethiopia has been implementing the 'Integrated Management of Childhood Illness' programme since 1997 later scaled up to the national level in 2007.[5] LRIs, and pneumonia in particular, have been among the top three leading causes of childhood mortality in the country.[6] Among LRIs' aetiologies, *Streptococcus pneumoniae* contributed to more deaths than the other LRIs aetiologies combined.[3] In 2011, the country introduced 10-valent pneumococcal conjugate vaccine (PCV 10) into its national immunisation programme to reduce the burden of Streptococcal pneumonia.[7]

Morbidity and mortality from LRIs are attributable to multiple underlying factors. Malnutrition is one of the main underlying risk factors.[8 9] The other main attributable factor are poor living conditions that include household crowding, parental smoking, high use of household solid fuel/biomass consumption, poor ventilation and lack of hand-washing facilities.[10–12] In addition, bottle feeding also contributes to the burden of LRIs in children.[13]

The flagship Ethiopian Health Extension Programme (HEP) has been the backbone of the country's health system strategies to reduce the burden of LRIs and other diseases through preventive and health promotion activities at the community level. The HEP has also improved broader access to healthcare, availability of essential antibiotics and immunisation mainly to the rural population since 2004.[14] In 2010, the country also introduced the 'integrated community case management' (ICCM) approach to treat pneumonia through trained health cadres of health extension workers (HEWs) implementing HEP.[15]

Currently, Ethiopia is implementing its Health Sector Transformation Plan-2 (HSTP-2) which is adapted from the Sustainable Development Goals. Some of the aims include increasing the proportion of under-5 children with pneumonia who received antibiotics from 48% to 69% and improving full vaccination coverage from 44% to 69% between 2020 and 2025.[16] Commitment to international goals such as The Global Action Plan for the Prevention and Control of Pneumonia and Diarrhoea by 2025 could be reached if enough investment is made in high LRI burden countries such as Ethiopia.[17]

As Ethiopia is a country of stark contrasts in socioeconomic, epidemiological and geographical variations, estimating disease burden at the regional level could provide valid and reliable information to inform policy decisions, including efficient resource allocation to match the burden in the subnational states. Hence, this article presents the 2019 Global Burden of Diseases, Injuries and Risk Factors study (GBD) results on the burden, trends and regional variations of LRIs in Ethiopia from 1990 to 2019.

## METHODS
### Study setting
Ethiopia is the second-most populous country in Africa next to Nigeria, with an estimated population of 112 million in 2019.[18] More than half of the country's population is under 20, and over 80% of the population resides in rural areas.[19] The country is subdivided into 10 regional states (Afar, Amhara, Benishangul-Gumuz, Gambella, Harari, Oromia, Somali, Sidama, Southern Nations and Nationalities and Peoples (SNNP), and Tigray) and two chartered cities (Addis Ababa and Dire Dawa). During this study, Sidama was a zonal administration under the SNNP region. Oromia, Amhara and SNNP are the highly populated regions. In this study, we classified the regions into urban (Addis Ababa, Dire Dawa and Harari), agrarian (Oromia, Amhara, SNNP, Tigray) and pastoralist (Benishangul-Gumuz, Afar, Gambella and Somali). The socioeconomy of the regions such as income per person, educational attainment and total fertility rate varies as measured in sociodemographic index (SDI)[20] (figure 1).

The healthcare system of the country is a three-tiered system consisting of primary, secondary and tertiary levels of healthcare delivery units with 21 154 functioning health facilities and 159 545 health workforce in 2019.[16] The primary healthcare unit (PHCU) consists of health posts (staffed by HEWs), health centres and primary hospitals. The secondary level of care consists of general hospitals and the tertiary level of care includes national referral hospitals which provide specialised services.[16]

### Data sources and analysis
The analysis and findings of LRIs presented in this analysis were produced by the Ethiopia Subnational Burden of Disease Initiative, a collaborative endeavour between the National Data Management Center for Health (NDMC) at the Ethiopian Public Health Institute (EPHI) and the

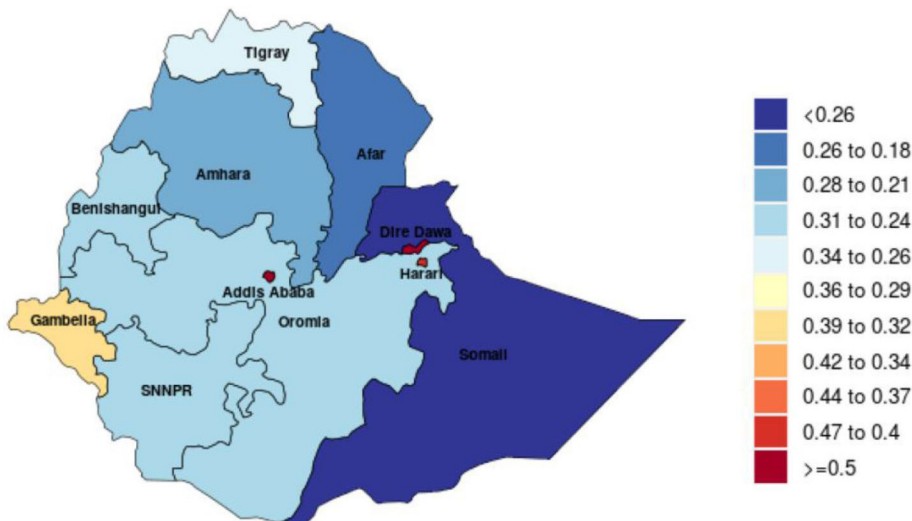

**Figure 1** Sociodemographic index, Ethiopia, 2019. SNNPR, Southern Nations and Nationalities and Peoples Region.

Institute for Health Metrics and Evaluation (IHME), as part of GBD. The details of the methodology were described elsewhere.[21] In brief, woreda (district) level geographical boundary mapping of regions and cities was used because woredas were relatively stable government structures (compared with lower or higher level administrative structures) during political or government changes and through the three census years (1984, 1994 and 2007). First, the analysis was estimated by mapping population and demography at the district level by time and region. Then, the data sources were mapped by regions before processing the data in the GBD analysis based on GBD protocol. EPHI, in collaboration with IHME, gathered all accessible data sources by location for Ethiopia and all regions and cities that included census, demographic surveillance, household surveys, diseases registry, health service utilisation, disease notification and other data for this analysis. A comprehensive description of data sources, quality and modelling for GBD 2019 has been reported on the following online portal: (http://ghdx.healthdata.org/gbd-2019/data-input-sources). This study outputs predates the COVID-19 and civil war occurred in the northern and other parts of the country and does not include the impact of COVID-19 or civil war.

### GBD methods and tools

The GBD details are reported elsewhere.[22] Diseases and injuries within the GBD were organised into levels: level 1 being the broadest causes of death and disability to level 4 being the most specific. Within the three level 1 causes (communicable, maternal, neonatal and nutritional diseases; non-communicable diseases; and injuries), there were 174 level 3 causes. The GBD 2019 study has estimated the burden of disease, including LRIs, for Ethiopia's national and subnational states. LRIs comprise diseases of the lower airways such as pneumonia, bronchitis and bronchiolitis, among others.[2] LRI mortality was estimated by age, sex, geography and year using a modelling platform called the Cause of Death Ensemble

model. LRI morbidity, including incidence, was modelled using a meta-regression platform known as DisMod-MR, a Bayesian, hierarchical, mixed-effects meta-regression platform.[23] Years of life lost (YLLs) were computed by multiplying cause-specific deaths by the life expectancy at the age of death.[24 25] Population risk assessments over time and among risks were estimated using the comparative risk assessment approach developed for the GBD study.[26 27] The GBD risk factors were categorised as follows: level 1 risk factors are behavioural, environmental, occupational and metabolic; level 2 risk factors include 20 clusters of risks; level 3 consists of 52 clusters of risks and level 4 contains 69 specific risk factors. All metrics were estimated separately for Ethiopia's nine regions and two chartered cities, and are presented with their 95% uncertainty intervals (UIs). All estimates produced for GBD report 95% UIs that account for sampling and non-sampling error associated with data and various assumptions of the modelling process and are derived from the 2.5th and 97.5th percentiles of 1000 draws.[22 28]

### Presentation of results

We present the burden of LRIs in Ethiopia and its regional states using incidence, deaths and YLLs categorised by sex, age groups and year. We used numbers, rates and per cent change for the quantification of the burden. We also estimated the risk factors contributing to LRIs in Ethiopia and the per cent change between 1990 and 2019. We reported GBD causes and risk factors using level 3 classifications, with 95% UIs. Additional tables and figures are attached in online supplemental materials.

### Patient and public involvement

Patients and the public were not involved in the design of the study.

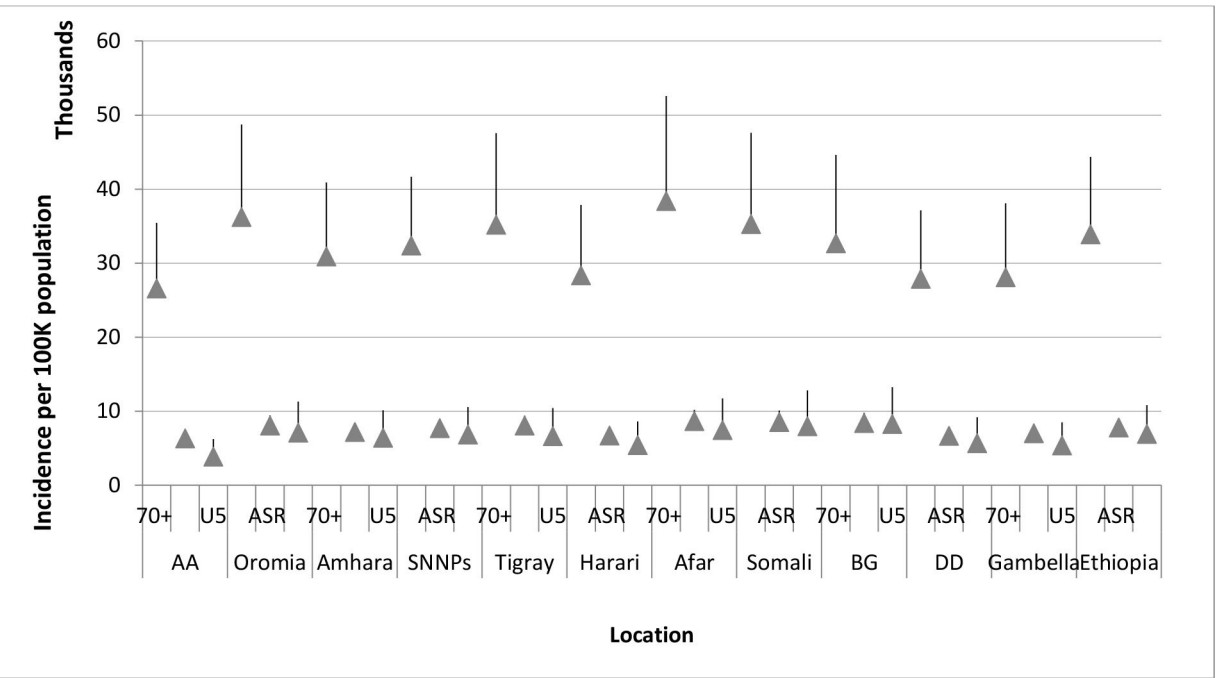

**Figure 2** Incidence of lower respiratory infections per 100 000 people in Ethiopia and its regions in 2019. AA, Addis Ababa; ASRM age-standardised rate; BG, Benishangul-Gumuz; DD, Dire Dawa; SNNPs, Southern Nations and Nationalities and Peoples.

## RESULTS

### Morbidity due to LRIs

In 2019, an estimated 6 628 673.6 (95% UI 6 108 786.2–7 230 986.3) new cases of LRIs occurred in Ethiopia resulting in an age-standardised incidence rate of 8313.7 per 100 000 people (95% UI 7757.6–8918). Out of the total LRI episodes, 22% (1 448 680.0 new cases (95% UI 1 150 089.8–1 799 704.4)) of new cases occurred among children younger than 5 years, yielding an annual incidence rate of 8685.0 per 100 000 children (95% UI 6895.1–10 789.8). In adults older than 70 years, there were 725 273.3 (95% UI 640 315.4–837 746.3) new cases of LRIs with an annual incidence of 38 394.4 per 100 000 people (95% UI 33 896.9–44 348.5) (figure 2 and online supplemental tables 1 and 2).

Compared with national estimates, a significantly lower age-standardised incidence rate of LRIs per 100 000 people was observed in the chartered cities (6788.1 (95% UI 6285.1–7339.1) in Addis Ababa, 7148.8 (95% UI 6634.6–7750.7) in Dire Dawa) and in Harari region (7190.5 (95% UI 6684.1–7718.6)). The highest rates of age-standardised incidence per 100 000 people were observed in Afar (9350.2 (95% UI 8648.8–10 157.4)), Somali (9220.0 (95% UI 8515.9–10 046.4)) and Benishangul-Gumuz (9054.6 (95% UI 8394.2–9766.0)) although not significant compared with the national estimate (figure 2 and online supplemental figure 1 and table 2).

In children younger than 5 years, the lowest incidence rates were observed in Addis Ababa (4927.2 (95% UI 3812.5–6231.8)), Harari (6821.6 (95% UI 5373.5–8607.3)) and Gambella (6791.1 (95% UI 5370.4–8460.1)) per 100 000 people, substantially below the national estimate. The highest incidence of LRIs per 100 000 children were recorded in Benishangul-Gumuz (10 481.8 (95% UI 8269.5–13 219.0)), Somali (10 062.7 (95% UI 7949.6–12 802.8)), Afar (9365.5 (95% UI 7421.7–11 714.7)) and Oromia (9039.4 (95% UI 7080.5–11 277.7)).

The age-standardised decline rate between 1990 and 2019 was 39% for both sexes, and it was 56% in under-5 children and 13% in adults older than 70. The lowest decline in age-standardised incidence rate was in Somali (19%); while it was between 38% and 44% for the remaining regions (online supplemental table 2).

In children younger than 5, the incidence rates increased slightly between 1990 and 1995, except in Addis Ababa and Amhara region. The highest decline rates were found between the years 2005 and 2015 across all regions (online supplemental figure 2).

### Mortality due to LRIs

In 2019, LRIs caused 46 300.7 (95% UI 39 515–54 642) deaths in Ethiopia, giving the age-standardised mortality rate of 86.4 (75.3–97.6) per 100 000 people. Under-5 mortality accounted for 42% (19 591.8 (95% UI 14 018.4–26 899.0)) of all deaths due to LRIs which resulted in a mortality rate of 117.4 (84.0–161.2) per 100 000 children. Of the under-5 deaths, 71% (13 919.0 (95% UI 9946.0–18 860.0)) occurred in the first year of life. In adults older than 70 years, LRIs caused 14 627.6 (95% UI 12 393.7–16 892.8) deaths, which was a mortality rate of 774.3 (95% UI 656–894.2) deaths per 100 000 adults (table 1 and online supplemental table 3).

**Table 1** Lower respiratory infections mortality rates and percentage changes between 1990 and 2019 in Ethiopia, both sexes, with different age groups

| Location | Age standardised | | Children younger than 5 | | | People older than 70 | | |
|---|---|---|---|---|---|---|---|---|
| | Deaths per 100 000 people (95% UI), 1990 | Deaths per 100 000 people (95% UI), 2019 | Deaths per 100 000 children (95% UI), 1990 | Deaths per 100 000 children (95% UI), 2019 | Change, % | Deaths per 100 000 people (95% UI), 1990 | Deaths per 100 000 people (95% UI), 2019 | Change, % |
| Addis Ababa | 163.4 (134.6–206.9) | 59.4 (49.8–71.4) | 441.5 (320.4–602.7) | 23.4 (14.1–36.4) | 95 | 909.2 (669.5–1264.6) | 503.6 (410.5–633.7) | 45 |
| Oromia | 241.3 (188.5–292.8) | 89.3 (75.9–103.1) | 920.7 (629.5–1264.1) | 122.3 (84.8–171) | 87 | 1174.1 (842.8–1543) | 849.5 (687.3–1009) | 28 |
| Amhara | 197.7 (163.5–236.5) | 74.3 (59.4–91) | 691.3 (531.7–883.4) | 94.9 (55.6–146.1) | 87 | 1017.9 (755.4–1324) | 682.7 (537.4–857.9) | 33 |
| SNNPs | 243.7 (196.4–296.9) | 98.9 (83.8–116.6) | 971.9 (697.2–1273.2) | 118.4 (79–168.5) | 88 | 1161.3 (584.7–822.3) | 845.4 (685.6–018.1) | 28 |
| Tigray | 242.7 (200.5–292) | 84.6 (69.2–100.1) | 780.4 (593.5–1009.1) | 67.2 (43.5–98.3) | 92 | 1166.7 (838.2–1586) | 788.6 (637.9–956.8) | 33 |
| Harari | 240.2 (182.6–303.1) | 77 (63–92.5) | 1146.7 (730–1604.1) | 89.9 (51.1–138) | 93 | 762.3 (412–1223.7) | 694.3 (551.9–849) | 9 |
| Afar | 244.8 (186.5–316.9) | 103.7 (86.6–122.6) | 723.7 (481.5–1023.7) | 101.7 (65–152.4) | 86 | 1010 (658–1541.7) | 917.6 (720.6–146.8) | 10 |
| Somali | 147.6 (113.8–192.6) | 97.5 (79.2–118.6) | 455.9 (313–628.5) | 197.3 (135–279.8) | 57 | 798.8 (519.2–1177.6) | 765.1 (588.8–972.4) | 5 |
| BG | 284.1 (221.8–358.1) | 101.8 (84–121.7) | 1266.4 (846.7–790.9) | 215.1 (141.4–311) | 84 | 975.3 (656.5–1379.3) | 671.9 (532.7–848.7) | 32 |
| Dire Dawa | 220.1 (171.6–270.9) | 69.9 (56.5–84) | 1084.2 (689.1–503.2) | 83.7 (44.7–135.9) | 93 | 868.7 (604.5–1205.5) | 621.3 (496.4–766.1) | 29 |
| Gambella | 231.5 (176.8–297.4) | 82.4 (68.3–97.1) | 1265.5 (805.4–764.3) | 57 (32.4–89.6) | 96 | 739.6 (477.7–1074.9) | 678.3 (540.4–835.3) | 9 |
| Ethiopia* | 223 (184.7–264.3) | 86.4 (75.3–97.6) | 822.2 (635–1051.2) | 117.4 (84–161.2) | 86 | 1092.3 (844–1391.5) | 774.3 (656–894.2) | 30 |

N.B: all changes are in decreasing per cent.

*National estimate.

BG, Benishangul-Gumuz; SNNPs, Southern Nations, Nationalities and Peoples; UI, uncertainty interval.

In 2019, a number of deaths in all age groups by region were highest in Oromia (18 206 (95% UI 15 193–21 745)), Amhara (9525 (95% UI 7530–11872)), SNNP (9494 (95% UI 7713–11 649)) and Somali (3907.5 (95% UI 3007.4–4959.4)), followed by Tigray (2551.4 (95% UI 2090.3–3028.6)), Afar (706.2 (95% UI 567.7–869.6)), Benishangul-Gumuz (619 (95% UI 470.7–803.0)), Dire Dawa (154.6 (95% UI 120.6–193.8)) and Gambella (123.2 (95% UI 98.2–151.9)). Harari (93.1 (95% UI 73.8–116.0) had the lowest number of deaths (online supplemental table 3).

Between 1990 and 2019, the age-standardised mortality rate declined by 61%, and the decline in under-5 children and adults over 70 years was 86% and 30%, respectively (table 1 and online supplemental figure 3).

In Ethiopia, the age-standardised mortality rate was higher among males (100.6 (95% UI 84.1–121.4)) than females (71.8 (95% UI 60.2–82.9)) (online supplemental table 4).

In 1990, the age-standardised mortality rate per 100 000 people was the highest in Benishangul-Gumuz (284.1 (95% UI 221.8–358.1)), Afar (244.8 (95% UI 186.5–316.9)) and SNNP (243.7 (95% UI 196.4–296.9)), although it is not significantly different from the national estimate. On the other hand, the lowest age standardised mortality rate was exhibited in Somali (147.6 (95% UI 113.8–192.6)). Although the value was not significantly different from the national value in 1990, Addis Ababa showed the second lowest age-standardised mortality rate (163.4 (95% UI 134.6–206.9)) (table 1).

In 2019, the age-standardised mortality rate per 100 000 people was significantly lower in Addis Ababa (59.4 (95% UI 49.8–71.4)) when compared with other regions, although not significantly lower than Gambella and other urban areas (Harari and Dire Dawa). Dire Dawa showed a significantly lower age-standardised mortality rate when compared with Benishangul-Gumuz, Afar and SNNP. Compared with the national estimate, the regions of Afar (103.7 (95% UI 86.6–122.6)) and Benishangul-Gumuz (101.8 (95% UI 84.0–121.7)) recorded the highest age-standardised mortality rates per 100 000 people, although the difference is not statistically significant (table 1). Compared with the 1990s, there was a 58%–68% decrease in the age-standardised mortality rates across all regions. However, the Somali region recorded a 34% reduction in mortality rates (online supplemental figure 3).

Among children below the age of 5, the mortality rate per 100 000 people was significantly lower in Addis Ababa (23.4 ((95% UI 14.1–36.4)) than other regions, although it was not significantly less than Gambella in 2019. The mortality rate was the highest in Benishangul-Gumuz (215.1 (95% UI 141.4–311.0)), Somali (197.3 (95% UI 135.2–279.8)) and Oromia (122.3 (95% UI 84.8–171.0)) despite being not significantly higher than the national estimate (table 1). Dire Dawa showed a significantly lower mortality rate than Benishangul-Gumuz and Somali. Harari had a significantly lower mortality rate than Benishangul-Gumuz. For all regions, the mortality

rate declined for children younger than 5 years between 1990 and 2019 by between 84% (in Benishangul-Gumuz) and 96% (in Gambella), except in Somali, which showed a 57% decline (table 1). The mortality rate increased slightly between 1990 and 1995 and between 2010 and 2015 in Somali region (online supplemental figure 4).

The mortality rate in adults older than 70 was the lowest in Addis Ababa (503.6 (95% UI 410.5–633.7)). Other regions and cities have not shown a statistically significant difference from the national estimate. The decline in mortality rates between 1990 and 2019 is below 50% across all regions (table 1). The mortality rate increased in Afar and Somali between 2005 and 2019 and in Gambella between 1990 and 2005 (online supplemental figure 5).

## Premature mortality due to LRIs

In 2019, premature death due to LRIs was 2 445 093.7 (95% UI 1 934 420.8–3 119 838.6) YLLs, yielding an age-standardised rate of 2404.5 per 100 000 people (95% UI 2059.4–2833.3). Compared with 1990, the age-standardised YLL rate declined by 76% in 2019. In parallel, 70% of all premature mortality occurred in children younger than 5, which accounted for 1 721 122.3 (95% UI 1 231 032.1–2 362 958.7) YLLs. The YLL rate of 72 055.4 (95% UI 55 718.5–92 064) per 100 000 under-5 children in 2019 declined by 86% compared with the YLL rate in 1990. Adults over 70 years contributed 8% of all YLLs due to LRIs (194 756.2 (95% UI 165 462.0–225 502.1)), yielding a rate of 10 309.9 (95% UI 8759.2–11 937.6) YLLs per 100 000 people (table 2; online supplemental tables 5 and 6).

Compared with 1990, the number of YLLs has decreased by 70% in all age groups and by 76% in children younger than 5 in 2019. However, the number of YLLs has increased by 45% in adults older than 70 (online supplemental table 5).

The age-standardised YLL rate was significantly lower in Addis Ababa (1285.6 (95% UI 1065–1561.8)) in 2019 compared with the national average. The highest age standardised YLL rate were in Benishangul-Gumuz (3571.1 (95% UI 2772.4–4510.7)), Somali (3236.4 (95% UI 2537.4–4006)) and Afar (2824.6 (95% UI 2323–3414)), although not statistically significant compared with the national estimate (table 2).

In children younger than 5 years, Addis Ababa (2069 (95% UI 1253–3217.5)) had a significantly lower YLL rate than the national estimate. The YLL rate was observed to be the lowest in Gambella (5031.8 (95% UI 2867.2–7885.7)), although not significantly lower than the national estimate (table 2).

The age-standardised premature mortality rate between 1990 and 2019 showed a continuous decline in all regions, except Somali. Benishangul-Gumuz showed the highest burden between 1990 and 2019 (figure 3).

The change in premature mortality in children younger than 5 was significant throughout the years. However, the reduction in premature mortality among adults older

**Table 2** Lower respiratory infections YLL rates and percentage changes between 1990 and 2019 in Ethiopia, both sexes, with different age groups

| Location | Age standardised | | Children younger than 5 years | | | People older than 70 | | |
|---|---|---|---|---|---|---|---|---|
| | YLL per 100000 people,1990 | YLL per 100000 people, 2019 | YLL per 100000 people,1990 | YLL per 100000 people, 2019 | Change, % | YLL per 100000 people,1990 | YLL per 100000 people, 2019 | Change, % |
| Addis Ababa | 6362.2 (5161.9–7847.9) | 1285.6 (1065–1561.8) | 38824.7 (28230.1–53092.4) | 2069 (1253–3217.5) | 95 | 13587 (9985.4–18978.3) | 6857.1 (5545.8–8721.2) | 50 |
| Oromia | 11217.6 (8517–14318) | 2433.8 (2042.2–2879.5) | 80610.7 (55103.7–110696) | 10742.8 (7456.7–15039.6) | 87 | 18754.2 (13297.1–24843) | 11119.7 (8939.1–13305) | 41 |
| Amhara | 8690.4 (7146.3–10453.8) | 2016.4 (1551.4–2541.8) | 60663.3 (46712.5–77511.5) | 8355 (4898.9–12845) | 87 | 15950.4 (11723.4–21059.7) | 9059.7 (7063.2–11529.2) | 44 |
| SNNPs | 11633 (9119.4–14477.9) | 2698.1 (2243.5–3250.2) | 85114.7 (61203.2–111502.1) | 10409.3 (6968.1–14819.3) | 88 | 18842.6 (13166.1–26081) | 11736.6 (9426.7–14238.6) | 38 |
| Tigray | 10346.4 (8522.9–12340.3) | 1977.1 (1593.9–2395) | 68509.3 (52185.7–88712.2) | 5927.9 (3839.4–8676.8) | 92 | 19502.8 (13914.1–26751.2) | 10486.8 (8439.5–12765.3) | 47 |
| Harari | 12826.5 (9336.5–16985.4) | 2060.4 (1595.6–2623.1) | 100421.7 (63898.7–140467.8) | 7922.5 (4518.7–12117.2) | 93 | 11782.6 (6360.4–19341.1) | 9260.4 (7338.7–11340.3) | 22 |
| Afar | 10993.5 (8520.8–14193.6) | 2824.6 (2323–3414) | 63570.5 (42296.6–89769) | 8943.7 (5734.6–13390.2) | 86 | 16925.5 (10923.7–26001.7) | 11867.1 (9346–14716.9) | 30 |
| Somali | 6286.9 (4860.5–7970) | 3236.6 (2537.4–4006) | 40088.1 (27543.7–55109) | 17314.3 (11890.7–24530) | 57 | 11953.6 (7599.2–17885.5) | 10246 (7797.8–13150.6) | 15 |
| BG | 14965.1 (11197.4–19998.6) | 3571.1 (2772.4–4510.7) | 110847.2 (74368–156588.4) | 18868.3 (12442.9–27281.8) | 83 | 16397.3 (10917–23228.5) | 9541.9 (7476.9–12112.3) | 42 |
| Dire Dawa | 11749.7 (8605.4–15249.5) | 1832.8 (1407.3–2363.1) | 94905.3 (60458.5–131584.5) | 7371.3 (3953–11962.1) | 93 | 13207.9 (9219.6–18328.2) | 8371 (6616.3–10408.2) | 37 |
| Gambella | 13175.8 (9291–17411) | 1937.4 (1567.6–2344.6) | 110783.8 (70576–154304.1) | 5031.8 (2867.2–7885.7) | 96 | 11811.4 (7673.5–17012.5) | 10030.9 (7730.2–12664.9) | 16 |
| Ethiopia* | 10189.1 (8347.5–12201.8) | 2404.5 (2059.4–2833.3) | 72055.4 (55718.5–92064) | 10318.6 (7380.4–14166.6) | 86 | 17415.9 (13362.7–22228.5) | 10309.9 (8759.2–11937.6) | 41 |

*National estimate.
BG, Benishangul-Gumuz; SNNPs, Southern Nations Nationalities and Peoples; YLL, year of life lost.

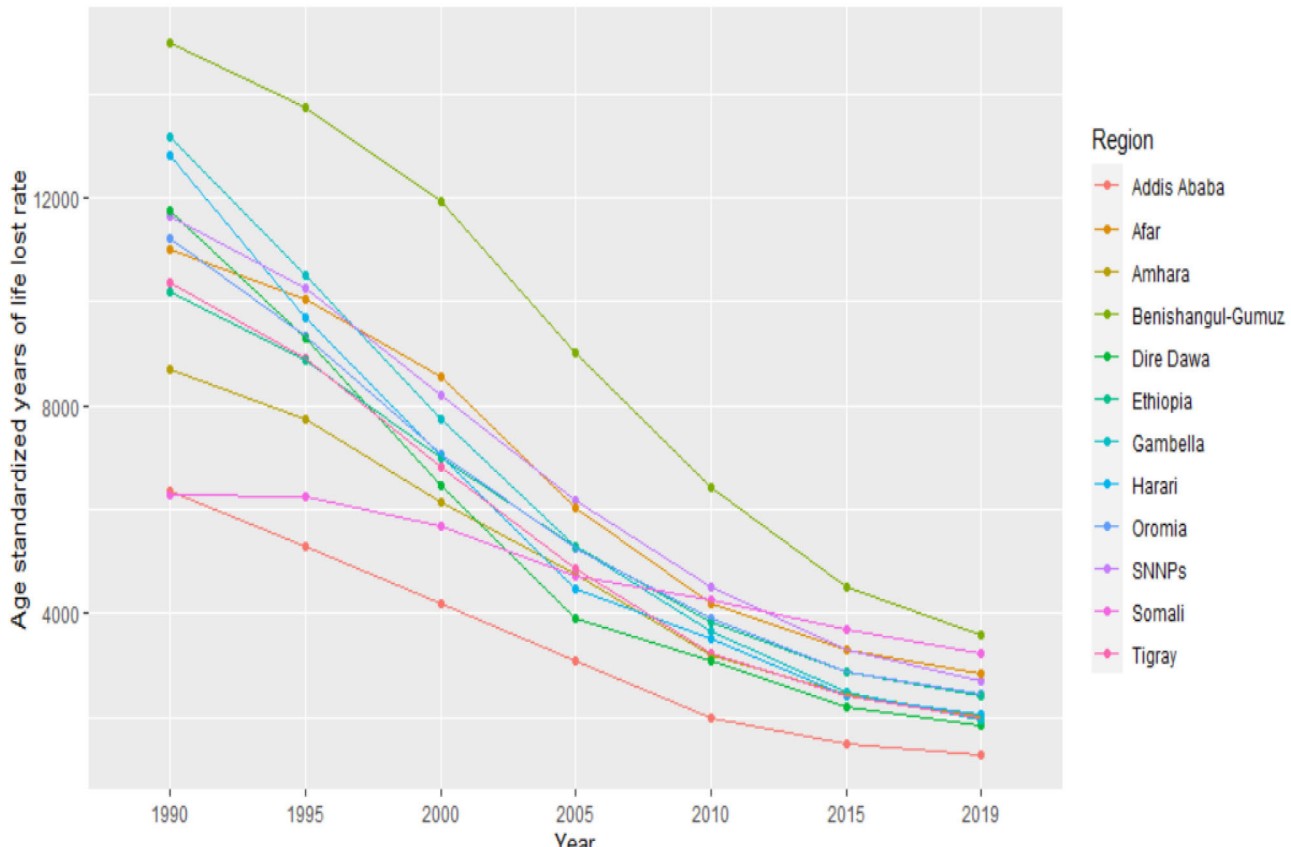

**Figure 3** Trend in LRIs age-standardised years of life lost rates per 100 000 people in Ethiopia, 1990–2019. LRIs, lower respiratory infections; SNNPs, Southern Nations and Nationalities and Peoples.

than 70 was not as significant (online supplemental figures 6 and 7).

### Risk factors

Across all population groups, about half (48%) of the mortalities (measured in rates) due to LRIs in Ethiopia were attributed to household air pollution from solid fuel. In addition, lack of access to a hand-washing facility (23%), childhood wasting (23%), low birth weight (9%), short gestation period (7%) and ambient particulate matter pollution (6%) were also risk factors for mortality due to LRIs. The contribution of the risk factors to death due to LRIs in all regions was similar to the national estimate, except in Addis Ababa, where lack of access to hand-washing facilities, ambient particulate matter pollution, and low temperature were the main contributing factors. Ambient particulate matter pollution was also relatively higher in Dire Dawa, Harari and Tigray regions (online supplemental figure 8).

In children younger than 5, more than three-fourths of deaths due to LRIs were attributed to childhood wasting (54%) stunting (12%) and child underweight (10%) in Ethiopia. In addition, 50% of the LRIs mortalities were attributed to household air pollution from solid fuel. Lack of access to hand-washing facilities (23%), low birth weight (23%), short gestation period (17%), high (4%) or low temperature (4%), absence of exclusive breast feeding (4%) were also the risk factors with evident

contribution. The distribution of risk factors varied among the regions. Child wasting, low birth weight and preterm birth were the contributing factors of mortality in Addis Ababa. In Harari and Dire Dawa, child wasting, lack of access to hand-washing facilities, preterm birth and household air pollution from solid fuels contributed more to the death from LRIs than the other risk factors in 2019. All the risk factors except for ambient particulate air matter and low temperature were the highest in Somali and Benishangul-Gumuz (online supplemental figure 9). When we examine the trend from 1990 to 2019, there is no significant reduction in the risk factors (online supplemental figure 10).

### Discussion

The findings from this study indicate that although the burden of LRIs, measured in incidence or mortality rates, has shown a significant decline, they are still the third-leading cause of death after neonatal disorders and diarrheal diseases in 2019 in Ethiopia. Cities and predominantly urban areas had lower mortality rates than predominantly pastoralist regions of the country. The rate of decline in mortality between 1990 and 2019 varied slightly across regions and chartered cities. The mortality rate decreased by more than three-fourths among children under the age of 5 and only by one-third among adults older than 70 between 1990 and 2019. Half of LRIs mortalities are attributed to household air pollution from

solid fuels in all age groups. About three-fourths of LRIs in children were attributed to malnutrition.

The mortality rate of LRIs among children below the age of 5 has declined by 86% between 1990 and 2019. This decline could be attributed to improvements in living conditions, access to healthcare and immunisation. The national health delivery infrastructure has grown from 2600 health facilities in 1997–21 154 facilities, including 314 hospitals, 3678 health centres, and 17 162 health posts and private health facilities in 2019. As a result, the health workforce has increased from 46 000 in 2007 to 159 545 in 2019.[16] The introduction of HEP since 2004 to provide preventive, health promotion and curative treatment for pneumonia, malaria and diarrhoea had improved health outcomes for children.[14] Moreover, Ethiopia has been implementing holistic child health improvement programmes like IMCI since 1997[15] and ICCM.[5] Vaccination against *S. pneumoniae*, which is responsible for about half of the LRIs mortality in African countries, with the PCV 10 vaccine since 2011 helped reduce the burden of LRIs.[4] In addition, improved socioeconomic conditions have supported the reduction of the burden of LRIs in the country.[29]

Despite the achievements reached in improving child health outcomes in Ethiopia, children are still disproportionately affected by LRIs. Out of the total burden of LRIs, 42% of all the deaths were among children younger than 5. In that regard, this study has shown that 1 out of 10 child deaths are due to LRIs. However, another study estimated that pneumonia alone shared about 17% of all deaths in children younger than 5 years.[30] Among children under 5, children younger than one carry the highest burden of LRIs. Furthermore, the mortality rate among children younger than one (397.7 per 100 000 people) was more than nine times higher than that of children between 1 and 4 years old (43 per 100 000 people). Although most studies on the high burden of LRIs corroborate our findings, the findings from Global Health Observatory (GHO) estimated a much higher mortality rate than the estimates of this study (ie, 481.9 in children less than 1 year and 51.1 per 1 000 000 population in children between 1 and 4 years).[8] Although we could not explain why this variation occurs, the estimates from GBD are also less than the estimate from findings of the Child Health Epidemiology Reference Group (CHERG), which is primarily due to the difference in the types of data used.[31]

Urban areas of the country, mainly Addis Ababa and Dire Dawa, had significantly lower mortality rates than pastoralist areas such as Benishangul-Gumuz and Somali. These regional variations could be attributed to gaps in availability and access to healthcare and socioeconomic status differences among the subnational states. A previously conducted study indicated that disease burden is high among people in the poorest wealth quintile and people located mainly in Afar, Somali, Oromia, SNNP and Benishangul-Gumuz and also have the lowest level of health service utilisation.[32] The rural area of the country has less healthcare coverage and utilisation than the urban areas. The coverage of all basic vaccination is 43% in the country. Across the regions, the coverage of all basic vaccination is lowest in Afar (20%) and highest in Addis Ababa (83%). Coverage of pentavalent vaccines in children is 72% among the urban population while it is 56% among the rural population.[15 30] There is a variation in the performance of immunisation across regions. Addis Ababa, Ethiopia's capital has PCV3 coverage of 93% among 1 year in 2019, whereas this coverage is only 23% in Afar and Somali. The wasting rate in Somali was 21% in children under 5, while it was just 2% in Addis Ababa in 2019[33]

In addition to children, LRIs affects people older than 70 years. The number of people dying from LRIs increased over the years in people older than 70 years, partly due to the increase in the ageing population. However, the mortality rate declined by 30% between 1990 and 2019.[3 34] The incidence rate did not show a significant improvement across the study years, showing only 15% reductions between 1990 and 2019. Among the regions, Somali and Afar have recorded an increased incidence rate between 1990 and 2019. This could partly be explained by poor accessibility and availability of health facilities in the regions.[33]

Wasting, stunting and underweight were major risk factors contributing to the death of children younger than 5 years due to LRIs.[35–37] More than 37% of children under 5 are stunted, with a higher percentage in rural areas (41%) than in urban areas (26%). Similarly, the prevalence of child underweight and wasting is 21% in the rural and 7% in the urban parts of Ethiopia.[38] This indicates that more investment is needed to reduce the burden on malnutrition among children and in the rural parts of the country to attain better health outcomes, protecting against LRIs. Although the prevalence of stunting, wasting and underweight has decreased markedly over time, they are still major risk factors for death caused by LRIs. In the Millennium Development Goals (MDG) era, between 1990 and 2015, about half of the deaths averted due to LRIs were attributed to improvement in the nutritional status of children (reduction in wasting and stunting).[15]

Ambient particulate matter and household air pollution from solid fuel use were the two essential components of air pollution. Household air pollution from solid fuel use the was the second leading risk factor for LRIs, and ambient particulate matter is the eight leading risk factor among the top ten risk factors for LRIs.[39–41] Our analysis also showed that there is a poor progress in the reductions of risk factors across the year, which shows there is a weak attempt in reducing the risk factors to prevent the population from LRIs. This indicates that improved use of electricity and natural gas for cooking and also appropriate investment in interventions that helps to reduce these risk factors will contribute to the reduction in the burden of the LRIs.[3]

To reduce the burden of LRIs, both national and global efforts are underway. The Global Action Plan for

the Prevention and Control of Pneumonia and Diarrhoea (GAPPD), established by the WHO, set goals in 2013 to reduce child LRI mortality rates to below 3 in 1000 live births and to reduce severe LRI incidence by 75% of the 2010 baseline by 2025. To achieve these goals, reaching 90% of children with full-dose vaccine coverage, 90% access to pneumonia treatment, 50% coverage of exclusive breastfeeding in the first 6 months and exclusive breastfeeding promotion were set as prerequisites.[17] However, a 6% average annual mortality reduction was recorded between 2000 and 2018 in Ethiopia. With this, the country can only reach the 2025 GAPPD target in 2035, ten years behind the target, according to the Maternal and Child Epidemiology Estimation Group (MICE) estimation.[30]

Ethiopia's major health sector strategic plan, the HSTP-2 for the year 2021–2024, aims to reduce the infant mortality rate to 35, the neonatal mortality rate to 21 and the children under-5 mortality rate to 43 per 1000 live births. Similarly, the 2030 Sustainable Development Goals (SDG) has set targets to reduce childhood mortality significantly. These targets can be achieved if the country implements high-impact priority curative and preventive interventions against LRIs.[42] To achieve this, concerted action to improve policies, increase investment, foster innovations and scale-up evidence-based interventions has paramount importance. Parallel to this, an estimated US$274 billion for health is required to achieve the health-related SDGs by 2030 in 67 low-income and middle-income countries, including Ethiopia. One of the strategies to mobilise the needed resources is to increase government expenditure by 15% and share the population's costs through taxes or insurances.[43]

This study is not without limitations; limitations in the GBD methods also apply to this study and limitations on Ethiopia subnational burden of disease that includes scarcity of quality data is published elsewhere.[28] When data on causes of death, morbidity or risk factors were not available for a particular regional state such as Afar or Somali, GBD modelled estimates use data from other locations and predictive covariates. Data sources such as household surveys have both sampling and non-sampling errors that account and led a wider 95% UIs which might compromise the accuracy of the findings and reduce the use of these findings for policy decision-making.[1] Causes of mortality data sources used were mainly from verbal autopsy and sibling history having recall bias, broader category of causes of death report or poor generalisability to regional states.[44] We used the best available data identified through an extensive collaboration effort and involving more than 700 leading researchers and policymakers from Ethiopia. The generation of estimates and their interpretation have benefited from intensive subnational review workshops and consultative meetings with domain experts.

## Conclusion

Despite the substantial reduction in morbidity and mortality at national and regional states, LRIs still remain one of the leading causes of the burden of disease in Ethiopia. Children and elders are still disproportionately affected by LRIs. The burden of illness and death due to LRIs varies across regional states in Ethiopia, with lower rates in cities and predominantly urban areas while predominantly pastoralist areas of the country have higher rates. Efforts should be made to tackle the major risk factors contributing to death by LRIs. Improving child nutrition, access to immunisation and curative health services, as well as universal electrification to reduce indoor air pollution will be very useful strategies to reduce deaths due to LRIs in Ethiopia. Furthermore, improvement in socioeconomic factors will also help to reduce LRI burden at national and regional levels. To reach the targets set at the national and international level, mobilising resources to health and improving the provision of health services to the community according to the needs of the regions is of paramount importance.

**Author affiliations**
[1]Ethiopian Public Health Institute, Addis Ababa, Ethiopia
[2]Department of Health Metrics Sciences, University of Washington, Seattle, Washington, USA
[3]Jigjiga University, Jigjiga, Ethiopia
[4]Department of Health Systems and Policy, Addis Ababa University College of Health Sciences, Addis Ababa, Ethiopia
[5]Ethiopian Health Insurance Service, Addis Ababa, Ethiopia
[6]Department of Global Public Health and Primary Care Medicine, University of Bergen, Bergen, Norway
[7]Addis Center for Ethics and Priority Setting, Addis Ababa University, Addis Ababa, Ethiopia
[8]Harvard T.H. Chan School of Public Health, Harvard University, Cambridge, Massachusetts, USA
[9]Department of Anesthesiology, Dilla University College of Health Sciences, Dilla, Ethiopia
[10]Bahir Dar University College of Medical and Health Sciences, Bahir Dar, Ethiopia
[11]St Paul's Hospital Millennium Medical College, Addis Ababa, Ethiopia
[12]Addis Ababa University College of Health Sciences, Addis Ababa, Ethiopia
[13]Arba Minch University, Arba Minch, Ethiopia
[14]Adult Health Nursing, Debre Markos University College of Health Science, Debremarkos, Ethiopia
[15]Public Health, Arba Minch University, Arba Minch, Ethiopia
[16]School of Public Health, Bahir Dar University College of Medical and Health Sciences, Bahir Dar, Ethiopia
[17]Madda Walabu University, Bale-Goba, Ethiopia
[18]Bahir Dar University, Bahir Dar, Ethiopia
[19]Department of Public Health, Wollo University, Dessie, Ethiopia
[20]Wollo University, Dessie, Ethiopia
[21]Mekelle University, Mekelle, Ethiopia
[22]Department of Pharmacy, Bahir Dar University, Bahir Dar, Ethiopia
[23]Department of Public Health, Samara University, Semera, Ethiopia
[24]University of Gondar, Gondar, Ethiopia
[25]Department of Medical Laboratory Sciences, Debre Tabor University, Debre Tabor, Ethiopia
[26]Department of Epidemiology and Biostatistics, Wollo University, Dessie, Ethiopia
[27]Wolaita Sodo University, Sodo, Ethiopia
[28]Haramaya University College of Health and Medical Sciences, Harar, Ethiopia
[29]University of Gondar Hospital, Gondar, Ethiopia
[30]Department of Public Health, Madda Walabu University, Balle-Goba, Ethiopia
[31]Department of Nursing, Dilla University College of Health Sciences, Dilla, Ethiopia
[32]Madda Walabu University, Robe, Ethiopia
[33]Dire Dawa University, Dire Dawa, Ethiopia
[34]Department of Medical Laboratory Sciences, Arba Minch University, Arba Minch, Ethiopia
[35]School of Public Health, University of Washington, Seattle, Washington, USA

**Contributors** Providing data or critical feedback on data sources: SMA, MMA, AA, DA, MAA, HA, MAB, TH, TMB, BBAB, DBG, AH, AM, MN, NBS, YS, SW, MW, AY, FG and YEY. Developing methods or computational machinery: SMA, TMB, AH, MN, NBS and AY. Providing critical feedback on methods or results: SMA, MMA, GTA, AA, DA, ZG, MAA, DA, TA, HA, MAB, TH, TMB, AY, SAB, BBAB, CC, FMD, AAD, LE, TE, ZF, DBG, MDG, MAG, MG, AZ, AH, GM, AM, MN, BS, BB, MS, NBS, DS, YS, BW, SW, MW, AY, FG and YEY. Drafting the work or revising is critically for important intellectual content: SMA, GTA, MAA, DA, NCB, MAB, TMB, SAB, BBAB, CC, TE, ZF, DBG, MDG, AZ, FAG, MAH, AH, STM, AM, MN, BS, BB, NBS, YS, DMT, BW, AW, AY and FG. Managing the estimation or publications process: SMA, AM, MN, NBS and AY. Guarantor: AY and AM.

**Funding** Bill & Melinda Gates Foundation has funded Ethiopia subnational burden of disease EPHI-IHME collaborative initiative.

**Disclaimer** The funder of this study had no role in study design, data collection, data analysis, data interpretation, or the writing of the report.

**Map disclaimer** The inclusion of any map (including the depiction of any boundaries therein), or of any geographic or locational reference, does not imply the expression of any opinion whatsoever on the part of BMJ concerning the legal status of any country, territory, jurisdiction or area or of its authorities. Any such expression remains solely that of the relevant source and is not endorsed by BMJ. Maps are provided without any warranty of any kind, either express or implied.

**Competing interests** None declared.

**Patient and public involvement** Patients and/or the public were not involved in the design, or conduct, or reporting, or dissemination plans of this research.

**Patient consent for publication** Not applicable.

**Ethics approval** This manuscript was produced as part of the GBD Collaborator Network and in accordance with the GBD Protocol.

**Provenance and peer review** Not commissioned; externally peer reviewed.

**Data availability statement** Data are available on reasonable request.

**ORCID iDs**
Amanuel Yigezu http://orcid.org/0000-0003-2792-2163
Awoke Misganaw http://orcid.org/0000-0002-3949-9457
Alemayehu Hailu http://orcid.org/0000-0003-4872-8036
Solomon Tessema Memirie http://orcid.org/0000-0003-3806-2453
Semagn Mekonnen Abate http://orcid.org/0000-0001-5661-8537
Dejen Tsegaye http://orcid.org/0000-0002-3285-3855
Mulusew Andualem Asemahagn http://orcid.org/0000-0002-0345-9437
Daniel Atlaw http://orcid.org/0000-0002-2968-4958
Melaku Ashagrie Belete http://orcid.org/0000-0002-0706-9903
Setognal Aychiluhm Birara http://orcid.org/0000-0001-7565-7515
Chuchu Churko http://orcid.org/0000-0002-9132-9471
Tahir Eyayu http://orcid.org/0000-0002-2041-7183
Zinabu Fentaw http://orcid.org/0000-0002-4577-868X
Getahun Molla http://orcid.org/0000-0002-9926-7797
Biniyam Sahiledengle http://orcid.org/0000-0002-1114-4849
Migbar Sibhat http://orcid.org/0000-0002-1240-8551
Yonatan Solomon http://orcid.org/0000-0002-7798-7267
Shambel Wedajo http://orcid.org/0000-0002-3679-301X
Yazachew Engida Yismaw http://orcid.org/0000-0002-1758-9232

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
