## [Reviewer comments · BMJ Open]

ARTICLE DETAILS

TITLE (PROVISIONAL)	The burden of lower respiratory infections and associated risk factors across regions in Ethiopia: A subnational analysis of the Global Burden of Diseases 2019 Study
AUTHORS	Yigezu, Amanuel; Misganaw, Awoke; Getnet, Fentabil; Berheto, Tezera; Walker, Ally; Zergaw, Ababi; Gobena, Firehiwot; Haile, Muluken; Hailu, Alemayehu; Memirie, Solomon Tessema; Tolosa, Dereje; Abate, Semagn Mekonnen; Molla Adane, Mesafint; Akalu, Gizachew; Aklilu, Addis; Tsegaye, Dejen; Gebru, Zeleke; Asemahagn, Mulusew; Atlaw, Daniel; Awoke, Tewachew; Abebe, Hunegnaw; Belete, Melaku; Hailemariam, Tekleberhan; Yirga, Alemeshet; Birara, Setognal; Bodicha, Belay; Churko, Chuchu; Demeke, Feleke; Desta, Abebaw; Ena, Lankamo; Eyayu, Tahir; Fentaw, Zinabu; Gargamo, Daniel; Gebrehiwot, Mesfin; Gebremichael, Mathewos; Getachew, Melaku; Molla, Getahun; Sahiledengle, Biniyam; Beyene, Bereket; Sibhat, Migbar; Sidamo, Negussie; Solomon, Damtew; Solomon, Yonatan; Wagaye, Birhanu; Wedajo, Shambel; Weldemariam, Melat; Yismaw, Yazachew; Naghavi, Moshen

VERSION 1 – REVIEW

REVIEWER	Nugent, KM Texas Tech University, Division of Pulmonary and Critical Care Medicine
REVIEW RETURNED	17-Jan-2023

GENERAL COMMENTS	1. These authors are reporting study which evaluated the incidence, death rate, and years of life lost due to lower respiratory tract infections in Ethiopia between 1990 and 2019. They determined that children under the age of 5 accounted 42% of the mortality and 70% of the years life lost. There was a significant reduction in the rates between 1990 and 2019. Important factors included malnutrition in children and air pollution in homes secondary to solid fuel use. 2. This study includes an extraordinary amount of information and reports quite clear differences between the year 1990 and 2019, ages, and regional locations. Clearly the quality of the information in this report depends on the quality of information collected in various regional health units. It is difficult for any reviewer to comment on that. However, the authors do provide 95% uncertainty intervals for their point estimates in the tables. It is unclear to me how often various risk factors which contribute to the outcomes lower respiratory tract infection. Appendix tables 6 and 7 are difficult to interpret and would benefit from explanatory footnotes.
---

	3. The study does document trends and improvement of outcomes. In addition, the study indicates regions in Ethiopia which would benefit from more health care focused on children. 4. In my view this report provides important information and identifies potential approaches to improving the health in Ethiopia.
--	--

REVIEWER	Nenna, Raffaella Sapienza University of Rome
REVIEW RETURNED	27-Jan-2023

GENERAL COMMENTS	The manuscript of Amanuel Yigezu et al. is very interesting and reports a profound analysis of causes of mortality and morbidity between 1990 and 2019 in Ethiopia. The analysis is clear and precise, the data is very numerous, the results are clear but sometimes difficult to read. Some parts should be streamlined by putting some results in the supplement. The authors also report an analysis of risk factors. It is not clear from the text how these RF have changed over the observation time. This could allow Authors to deduce the impact of these RF in reducing mortality from LRTIs. Otherwise, the discussion on the causes of mortality reduction remains purely speculative.
---

REVIEWER	Hurley, James Ballarat Health Services
REVIEW RETURNED	21-Mar-2023

GENERAL COMMENTS	the burden and trends of morbidity and mortality due to LRIs, and the Risk Factors across Regions in Ethiopia, and the disparity across administrative regions and cities from 1990 to 2019. Abstract "This analysis is part of a collaborative and comparable systematic Global Burden of Disease (GBD 2019) study. " is unclear what is meant here, comparable to what; collaborative with who? It may be worth stating the obvious – this study predates COVID. Line 217 "Patients and the public were not involved.." I think you mean "...in the design of the study." Are some of the references incomplete? E.g. ref 18, 19, 28, 30, 31? Ref 20 What is "English' at the end of ref? Figure 1 y axis is unclear - 'thousands' is misaligned. Appendix tables 1 & S1 – S3, S5 - what is '95% UI'? These are presumably 95% Uncertainty Intervals (UI) – mentioned at line 214 but how these were calculated is not specified. Please add a statistical methods section to specify what and how the analysis was done. Is 'Presentation of results' self evident ? Tables 2 & s4 what is range? The limitations paragraph is too brief. If incompleteness of the data is an issue – how much is it an issue, how much and which data in particular? For example smoking prevalence? If vaccination has impacted LRI – how does this relate to prevalence of vaccination over the time period?
---

REVIEWER	Wagenhäuser , Isabell University Hospital Wurzburg
REVIEW RETURNED	22-Mar-2023

GENERAL COMMENTS	The authors report on the long-term incidence, mortality rate and reduction in life expectancy due to lower respiratory tract infections
--

in Ethiopia. In addition to the temporal development, regional correlations and factors influencing the parameters are also shown. Thus, important health policy and development goal-related health data are evaluated, which represent a relevant basis for further measures.

Major Comments:

Table 1 and Table 2: I think the percentage of change is very useful. Perhaps the use of arrows or +/- symbols to indicate the trend and quantification e.g. +, ++ and +++ would be useful to enable better readability as the table contains a lot of data.

The use of commas to separate the decimals per 1000 steps is not uniform. Since the numbers mentioned are sometimes very high, I would recommend, for the sake of readability, to always use commas consistently in steps of 1000 (e.g. in line 313 "3,571.1 [2,772.4-4,510.7]" instead of "3571.1 [2772.4-4510.7]").

In addition, due to the size of the numbers, I would omit the decimal place when specifying the YLL.

The discussion reads very long and narrative, perhaps it would be possible to shorten it a little and make it more precise.

Figure 1: The labelling of the y-axis is unclear, also the axis labels on the x-axis are difficult to assign. The use of colors, as in Figure 2, would also be useful. What statistical measure do the whiskers indicate?

It should be emphasised more clearly in the discussion that the data was collected before the COVID-19 pandemic and therefore may not be completely transferable to the current situation with COVID-19.

The limitations are explained very briefly and should be considered in a clearer and more differentiated way.

Lin 445: Missing source on the limitations of GBS in general. A short summary of the limitations in general in 1 - 2 sentences would also be useful for non-specialist readers.

In the Methods, a map of Ethiopia with the selected regions would be very useful, also for geographical classification in the African continent. For the Figures, it might also be useful to show these on several maps, with one map per measurement point, instead of just time courses.

Minor Comments:

Line 73: The abbreviation LRI is not introduced in the abstract.

Line 116: Is there a current source with concrete numbers after 2016?

Line 176: Maybe the wording "analysis" might be more scientific than "paper".

Line 225: Missing space before "38,394.4".

Line 301 (and elsewhere): Personally, I would prefer the phrase individual instead of people.

Line 360: Numbers up to and including twelve should not be written out numerically but as words, so instead of "the age of 5", for example, "the age of five".

Figure 2, 3 and 4: "Age-standardized" instead of "Age standardized" in the labelling of the x-axis.

VERSION 1 – AUTHOR RESPONSE

Reviewer: 1

Dr. KM Nugent, Texas Tech University

Comments to the Author:

1. These authors are reporting study which evaluated the incidence, death rate, and years of life lost due to lower respiratory tract infections in Ethiopia between 1990 and 2019. They determined that children under the age of 5 accounted 42% of the mortality and 70% of the years life lost. There was a significant reduction in the rates between 1990 and 2019. Important factors included malnutrition in children and air pollution in homes secondary to solid fuel use.

Authors' response: We thank the reviewer for the comment on our manuscript.

2. This study includes an extraordinary amount of information and reports quite clear differences between the year 1990 and 2019, ages, and regional locations. Clearly the quality of the information in this report depends on the quality of information collected in various regional health units. It is difficult for any reviewer to comment on that. However, the authors do provide 95% uncertainty intervals for their point estimates in the tables. It is unclear to me how often various risk factors which contribute to the outcomes lower respiratory tract infection.

Authors' response: we thank the reviewer for the important comment. The global burden of disease study estimates the attribution of risk factors to morbidity or mortality at a population level per 100,000. As we described in the methods section and cited references, population risk assessments over time and among risks were estimated using the comparative risk assessment approach developed for the GBD study. The comparative risk assessment methods used for estimating aggregate disease burdens attributable to different risk factors.

Appendix tables 6 and 7 are difficult to interpret and would benefit from explanatory footnotes.

Authors' response: we thank the reviewer for the important comment and we have now added a footnote to the appendix table 6.

3. The study does document trends and improvement of outcomes. In addition, the study indicates regions in Ethiopia which would benefit from more health care focused on children.

Authors' response: We thank the reviewer for the comment on our manuscript.

4. In my view this report provides important information and identifies potential approaches to improving the health in Ethiopia.

Authors' response: We thank the reviewer for the comment on our manuscript.

Reviewer: 2

Dr. Raffaella Nenna, Sapienza University of Rome

Comments to the Author:

The manuscript of Amanuel Yigezu et al. is very interesting and reports a profound analysis of causes of mortality and morbidity between 1990 and 2019 in Ethiopia. The analysis is clear and precise, the data is very numerous, the results are clear but sometimes difficult to read. Some parts should be streamlined by putting some results in the supplement.

Authors' response: We thank the reviewer for the comment on our manuscript.

The authors also report an analysis of risk factors. It is not clear from the text how these RF have changed over the observation time. This could allow Authors to deduce the impact of these RF in reducing mortality from LRTIs. Otherwise, the discussion on the causes of mortality reduction remains purely speculative.

Authors' response: We thank the reviewer for this important comment. We have now included the trend in the risk factors over the year. "When we examine the trend from 1990 to 2019, there is no significant reduction in the risk factors (Appendix Figure 8) (line 377-378)". "Our analysis also showed that there is a poor progress in the reductions of risk factors across the year, which shows there is a weak attempt in reducing the risk factors to prevent the population from LRIs. For example, the findings may indicate that improved use of electricity and natural gas for cooking and also appropriate investment in interventions that helps to reduce these risk factors to contribute to the reduction in the burden of the LRIs (line 462-468)".

Reviewer: 3

Dr. James Hurley, Ballarat Health Services, University of Melbourne

Comments to the Author:

the burden and trends of morbidity and mortality due to LRIs, and the Risk Factors across Regions in Ethiopia, and the disparity across administrative regions and cities from 1990 to 2019.

Abstract "This analysis is part of a collaborative and comparable systematic Global Burden of Disease (GBD 2019) study.

" is unclear what is meant here, comparable to what; collaborative with who?

Authors' response: We thank the reviewer for the comment. We have now modified the abstract design section as "This analysis used Global Burden of Disease 2019 framework to estimate morbidity and mortality outcomes of lower respiratory infection and its contributing risk factors. The Global Burden of Disease study uses all available data sources and Cause of Death Ensemble model to estimate deaths from lower respiratory infection and a Meta-Regression Disease Modeling technique to estimate lower respiratory infection non-fatal outcomes with 95% uncertainty intervals" (line 78-84)

It may be worth stating the obvious – this study predates COVID.

Authors' response: We have now included this section in the manuscript. "This study outputs predates the COVID-19 and civil war occurred in the northern and other parts of the country and does not include the impact of COVID-19 or civil war" (line 197-198).

Line 217 "Patients and the public were not involved.." I think you mean "..in the design of the study."

Authors' response: thank you, we have now modified the sentence. "Patients and the public were not involved in the design of the study". Line 251

Are some of the references incomplete? E.g. ref 18, 19, 28, 30, 31?

Ref 20 What is "English" at the end of ref?

Authors' response: we thank the reviewer for the important comment. We have now corrected the references.

Figure 1 y axis is unclear - 'thousands' is misaligned.

Authors' response: we thank the reviewer for the important comment. We have now corrected the figure 1.

Appendix tables 1 & S1 – S3, S5 - what is '95% UI'? These are presumably 95% Uncertainty Intervals (UI) – mentioned at line 214 but how these were calculated is not specified. Please add a statistical methods section to specify what and how the analysis was done. Is 'Presentation of results' self evident? Tables 2 & s4 what is range?

Authors' response: thank you for your comment. We have described different models in the method section and included this following statement to make clear how 95% UI was calculated, "All metrics using these models were estimated separately for Ethiopia's nine regions and two chartered cities, and are presented with their 95% uncertainty intervals (UIs). All estimates produced for GBD report 95% uncertainty intervals (UIs) that account for sampling and non-sampling error associated with data and various assumptions of the modelling process and are derived from the 2.5th and 97.5th percentiles of 1000 draws" (line 236-242).

The limitations paragraph is too brief. If incompleteness of the data is an issue – how much is it an issue, how much and which data in particular? For example smoking prevalence?

Authors' response: Thank you again for this important comment. We have revised and expanded this section in the text "limitations on Ethiopia subnational burden of disease that includes scarcity of quality data is published elsewhere (28). When data on causes of death, morbidity or risk factors were not available for a particular regional state such as Afar or Somali, GBD modelled estimates use data from other locations and predictive covariates such as health care access, income, education and others. Data sources such as household surveys have both sampling and non-sampling errors that account and led a wider 95% uncertainty intervals which might compromise the accuracy of the findings and reduce the use of these findings for policy decision-making (1). Causes of mortality data sources used were mainly from verbal autopsy and sibling history having recall bias, broader category of causes of death report or poor generalizability to regional states. We used the best available data identified through an extensive collaboration effort and involving more than 700 leading researchers and policy makers from Ethiopia. The generation of estimates and their interpretation have benefited from intensive subnational review workshops and consultative meetings with domain experts.

If vaccination has impacted LRI – how does this relate to prevalence of vaccination over the time period?

Authors' response: we thank the reviewer for the comment. Health care access such as vaccine coverage was used as covariates for estimating LRI related mortality and morbidity outcomes. As we have discussed, one of the explanation for improved outcome in LRIs could be increasing trends of vaccine overtime. However, the scope of our study does not include the impact of vaccination on reduction of LRI.

Reviewer: 4

Dr. Isabell Wagenhäuser , University Hospital Wurzburg

Comments to the Author:

The authors report on the long-term incidence, mortality rate and reduction in life expectancy due to lower respiratory tract infections in Ethiopia. In addition to the temporal development, regional correlations and factors influencing the parameters are also shown. Thus, important health policy and development goal-related health data are evaluated, which represent a relevant basis for further measures.

Authors' response: We thank the reviewer for the comment on our manuscript.

Major Comments:

Table 1 and Table 2: I think the percentage of change is very useful. Perhaps the use of arrows or +/- symbols to indicate the trend and quantification e.g. +, ++ and +++ would be useful to enable better readability as the table contains a lot of data.

Authors' response: we thank the reviewer for the important comments. We acknowledge the comment from the reviewer but we discussed to use the numbers as they are to be more specific on the values. The use of commas to separate the decimals per 1000 steps is not uniform. Since the numbers mentioned are sometimes very high, I would recommend, for the sake of readability, to always use commas consistently in steps of 1000 (e.g. in line 313 "3,571.1 [2,772.4-4,510.7]" instead of "3571.1 [2772.4-4510.7]"). In addition, due to the size of the numbers, I would omit the decimal place when specifying the YLL.

Authors' response: we thank the reviewer for the important comment. We have now used commas consistently between numbers in our manuscripts accordingly.

The discussion reads very long and narrative, perhaps it would be possible to shorten it a little and make it more precise.

Authors' response: Thank you for the important comment. We have now attempted to shorten the discussion section.

Figure1: The labeling of the y-axis is unclear, also the axis labels on the x-axis are difficult to assign. The use of colors, as in Figure 2, would also be useful. What statistical measure do the whiskers indicate?

Authors' response: We have now worked on figure 1 and adjusted the labels on the axis.

It should be emphasised more clearly in the discussion that the data was collected before the COVID-19 pandemic and therefore may not be completely transferable to the current situation with COVID-19.

Authors' response: thank you for the important comment. We have now included a section informing this study predates COVID-19. "This study outputs predates the COVID-19 and civil war occurred in the northern and other parts of the country and does not include the impact of COVID-19 or civil war" (line 217-219).

The limitations are explained very briefly and should be considered in a clearer and more differentiated way. Line 445: Missing source on the limitations of GBS in general. A short summary of the limitations in general in 1 - 2 sentences would also be useful for non-specialist readers.

Authors' response: Thank you again for this important comment. We have revised and expanded this section in the text "limitations on Ethiopia subnational burden of disease that includes scarcity of quality data is published elsewhere (28). When data on causes of death, morbidity or risk factors were not available for a particular regional state such as Afar or Somali, GBD modelled estimates use data from other locations and predictive covariates such as health care access, income, education and others. Data sources such as household surveys have both sampling and non-sampling errors that account and led a wider 95% uncertainty intervals which might compromise the accuracy of the findings and reduce the use of these findings for policy decision-making (1). Causes of mortality data sources used were mainly from verbal autopsy and sibling history having recall bias, broader category of causes of death report or poor generalizability to regional states. We used the best available data identified through an extensive collaboration effort and involving more than 700 leading researchers and policy makers from Ethiopia. The generation of estimates and their interpretation have benefited from intensive subnational review workshops and consultative meetings with domain experts. In the Methods, a map of Ethiopia with the selected regions would be very useful, also for geographical classification in the African continent. For the Figures, it might also be useful to show these on several maps, with one map per measurement point, instead of just time courses.

Authors' response: Thank you again, we have included a map in the methodology that shows socio-economic disparity between regional states. Figure 1

Minor Comments:

Line 73: The abbreviation LRI is not introduced in the abstract.

Authors' response: thank you for important comment. We have now introduced the abbreviation LRI in the abstract.

Line 116: Is there a current source with concrete numbers after 2016?

Authors' response: we thank the reviewer for this comment. However, we could not find a more concrete source after 2016.

Line 176: Maybe the wording "analysis" might be more scientific than "paper".

Authors' response: we thank the reviewer for the comment. We have now changed it to "analysis".

Authors' response: thank you for the important comment. We have now added space before the number.

Line 301 (and elsewhere): Personally, I would prefer the phrase individual instead of people.

Authors' response: we thank the reviewer for the comment. However, we discussed to keep it as it is.

Line 360: Numbers up to and including twelve should not be written out numerically but as words, so instead of "the age of 5", for example, "the age of five".

Authors' response: Thank you. We have modified such phrases.

Figure 2, 3 and 4: "Age-standardized" instead of "Age standardized" in the labelling of the x-axis.
Authors' response: we thank the reviewer for the comment. We have now adjusted the figures.